# Are We Still a Sexist Society? Primary Socialisation and Adherence to Gender Roles in Childhood

**DOI:** 10.3390/ijerph19063408

**Published:** 2022-03-14

**Authors:** Loredana Cerbara, Giulia Ciancimino, Antonio Tintori

**Affiliations:** Institute for Research on Population and Social Policies of the National Research Council of Italy (CNR-IRPPS), 00185 Rome, Italy; giulia.ciancimino@irpps.cnr.it (G.C.); antonio.tintori@cnr.it (A.T.)

**Keywords:** gender roles, primary socialisation, stereotypes, children, survey, sexism

## Abstract

Background: The internalisation of gender stereotypes has long-term impacts on the aspirations, opportunities and psychosocial well-being of people. The main objective of this study is to measure the adherence to gender roles among children, analysing the link between their roles’ internalisation, the family context and the socioeconomic environment. Method: During the Spring 2021, a survey was carried in Rome on children aged 8–11 through a structured questionnaire. The explanatory dimensions of the analysed topics were identified and a survey questionnaire with an ad hoc administration method were developed. Results: The results show a widespread internalisation of traditional gender roles among the respondents and differences by sex were found, since their acceptance is higher among boys for male roles and among girls for female roles. As the age increases, the adherence to male roles decreases for both boys and girls, while high levels of prosociality resulted in a lower adherence to female roles among boys. No significant relations were found with family and environmental variables. Conclusions: These findings show how the internalisation of gender stereotypes is already traceable at this age, and due to a different path of primary socialisation, boys and girls develop their gender identity consistent with social expectations. The lack of significant relations with environmental variables could be related to the age of the respondents, as the process of primary socialisation imbued with gender stereotypes still does not overlap secondary socialisation. These trends should be monitored during late childhood since at this age children are cognitively plastic, but also vulnerable and influenceable by surrounding stimuli. This research approach, especially if extended to a wider geographical scale, can provide important knowledge to support the relational well-being of children and equal opportunities of society as a whole.

## 1. Introduction

The idea that there are roles and activities appropriate only for males or females is acquired by children from the first years of life through the process of socialisation [1]. During this process, children build categories or schemes through which they interpret information from their immediate environment and develop their own gender identity, which evolve during their entire life [2]. Since the development of these schemes is strongly conditioned by the environment in which children live and from which they learn, during childhood, the family context and in particular parental communication and behaviour play a central role in the internalisation of stereotypes [3]. As age increases, especially during the primary school years, together with the family context, the socio-cultural context, teachers, peers and the media play a crucial role in the acquisition of social constraints that have long-term impacts on their aspirations and the perception of their opportunities, affecting also their relational and psychological well-being [4,5,6,7,8].

As is well known, the persistence of traditional gender roles exacerbates gender inequalities, prescribing rigid behaviours and unbalanced power relations within societies in favour of men [9,10]. According to the World Economic Forum, we will have to wait 135.6 years to achieve gender equality since the gap between women and men has increased by one more generation due to the COVID-19 crisis [11]. Numerous studies, already after the first months of COVID-19 spread and the consequent adoption of physical distancing measures, have detected an intensification of gender disparities at work and at home [12,13,14].

Although Italy has seen an improvement in terms of women’s political participation, it is the last country in Western Europe regarding women’s economic participation. The already existing high gender gap in the Italian labour market and the adherence to traditional gender roles, especially to those related to the domestic sphere of care and assistance, have been exacerbated by the pandemic [15,16,17].

The roots of the prejudices, discriminations and violence against women can be traced back to the still widespread gender stereotypes [18,19]. The persuasive power of stereotypes derives from the fact that they offer the opportunity to simplify, through schematisation, the complexity of information deriving from the external world, operating by generalisations and minimising intra-categorical differences [20]. Therefore, these social constrains represent real cognitive shortcuts that hinder social changes by strengthening the status quo, which consists of male supremacy [21]. Future citizens have the responsibility to build a society not permeated by gender stereotypes, which crystallise characteristics, skills, preferences and expected behaviours of men and women, limiting free initiative and the ability to face the future with flexibility and resilience.

Since the 1970s, studies on the adherence to gender stereotypes and roles have been increasing, investigating different aspects of the ways of their internalisation but also on their outcomes [22,23,24,25]. With reference to the population aged between 8 and 11 years old, which coincides with the end of childhood and the beginning of preadolescence, small groups have often been analysed, with the limit of not allowing inferential considerations on the target population [26,27,28]. Measuring the spread of gender roles among childhood and identifying individual and environmental factors that contribute to their strengthening is extremely important to offer children a chance of empowerment, which would allow the cultural evolution of our society. In fact, according to many developmental studies, during primary school, the rigidity of the gender categorisation decreases in favour of an increase in cognitive flexibility, also in relation to the adherence to gender stereotypes [29,30]. Thus, environmental influences become increasingly important for children’s acceptance of rigid gender roles, determining the meanings of the identity of themselves and others. At this regard, previous studies have analysed the variations in gender role attitudes and beliefs among children with different family backgrounds in terms of parental education and occupational status with conflicting results. Indeed, on the one hand, some research has highlighted the positive relation between the presence of gender stereotypes in children and medium and high parental education levels [31]. In contrast, other studies have found that, as the parental level of education increases, the adherence to traditional gender roles decreases [32,33,34,35].

Finally, the adoption of prosocial behaviours has a link with the adherence to gender roles. This relation can be explained considering the different path of primary socialisation between girls and boys. In fact, a stereotyped parental approach promotes more prosocial and affiliative behaviours in girls and more competitive and assertive behaviours in boys [36].

The development of prosocial behaviour starts during the first years of life, strengthening over time as a result of the socialisation process, which determine the diversification of behavioural paths for males and females [37]. Previous studies have pointed out that primary socialisation imbued with gender stereotypes promotes a greater prosociality for females [38]. Girls are usually more prosocial than boys, since from birth they learn that as females they should support and care for others by being nice and kind. The development of prosocial behaviour in boys can be developed with less adherence to female roles, as they learn from experience that care and assistance can be gender neutral [39].

Our study, which is based on a survey conducted on primary school children in Rome in 2021, fits into this framework with the main objective of measuring and analysing the adherence to gender roles among children aged 8–11 and the link among the level of gender roles’ internalisation and other main individual and contextual variables, as sex, age, the socio-economic context in which schools were located, the family context in terms of parental educational level and occupational status, the prosocial tendency and the time spent playing videogames or using social media. Indeed, to achieve real social change, the study of children’s adherence to gender roles and the identification of the most influential variables on the process of their internalisation are the keystone to provide families and schools with effective intervention tools, as well as to design public policies useful to counteract the spread of stereotypes, prejudices and violence.

Since the 1970s, the instruments to measure gender roles attitudes and stereotypes have proliferated within the scientific panorama. Starting from the pioneering works of Bem and Spence, respectively the Bem Sex Role Inventory and the Personal Attributes Questionnaire of Spence [40,41], many scales have been developed to measure traits of masculinity and femininity and the gender attitudes, such as the Sex Role Egalitarianism Scale [42], the Modern Sexism Scale [43], the Ambivalent Sexism Inventory [44] and the Attitudes Toward Women Scales for Adolescents [45]. The reason of their proliferation could be traced in the multidimensionality of gender roles attitudes and stereotypes and on their constant changes over time, across cultures and among social groups. The tool adopted in this paper follows the suggestions of the work of Liben and Bigler [46], which differentiates the measurement scales for age group developing a proper scale for children (the Children’s Occupational Activity Trait, COAT). This scale assesses the gender attitudes of children by asking them who can or who should perform certain occupations and activities, with the option of assigning items to both sexes rather than just to one of the sexes.

Starting from recent research on the positive impact of education on gender role attitudes [47,48,49], we hypothesised that: boys are more imbued with stereotypes concerning male roles and girls with those of female roles; the endorsement of these social conditioning decreases as the age of the respondents increases; with a high level of parental education and a more advantageous socioeconomic context, the level of adherence to the traditional gender roles among children decreases; the time spent in front of a screen is positive related to the internalisation of gender stereotypes; the tendency to prosocial behaviour inhibits the adherence to gender roles among boys, but not among girls.

## 2. Methodology

Between April and May 2021, a survey on children who were attending the last three years of primary school was carried out. As in Italy the duration of primary school is five years, the children involved in the survey were aged between 8 and 11 years. In total, eight schools in Rome were involved in the study, four belonging to the 6th district and the other four to the 8th districts. The city of Rome is divided into fifteen districts, which are called municipi. Each of these districts has different territorial extensions and population densities. The two selected areas represent, respectively, 9% and 5% of the resident population of Rome. The 6th and 8th districts are characterised by similar population density (between 2000 and 3000 inhabitants per km^2^) and heterogenous demographic and economic profiles. Together they have almost 400,000 inhabitants, that is, the equivalent to the population of a large city. The decision to involve these two districts of Rome responded to the need of observing any variations among different socio-economic contexts. In each selected school, 3 classes of student were randomly selected obtaining a cluster sampling selecting one class for the third year, one for the fourth and one for the fifth year. In this way, a two-stage stratified sample was obtained with stratification of the first stage units (schools) and random selection of the second stage units (clusters corresponding to classes of students). This sample can be considered probabilistic because the first level units were randomly selected and because the classes were selected randomly (but more freedom was given to school administrators to avoid problems in carrying out teaching). The representativeness of the sample can be associated with the districts, but not with the entire national panorama. The choice of involving two contexts different for their socio-economic characteristics allows to consider the result reliable for the formulation of hypotheses on similar contexts. The survey was carried out during the COVID-19 pandemic with its obvious obstacles in terms of the physical access of researchers to the classrooms. These obstacles were overcome thanks to the sensibility of the school administrations that recognised the importance and the urgency of the research. The researchers’ physical presence in the classrooms during the administration phase allowed to collect more reliable data, as their assistance guarantees the maximum understanding of the questions by the respondents as well as the minimisation of the teachers’ interference in the answers. In total, 412 interviews were collected, through a structured paper questionnaire consisting of 42 questions. For the administration phase, due to the young age of respondents, special attention was paid to the use of a suitable font type and size. The final sample was composed of 46.3% females and 53.7% of males. A total of 35.4% were attending the 3rd year of primary school, 31.7% the 4th year and 32.9% the 5th year. With the aim to obtain the parental consent in interviewing their children, a consent form was administered to each parent, with a short questionnaire on socio-demographic information about the respondents’ families. These socio-demographic forms were associated with the children’s questionnaires through a numeric code, in order to guarantee the anonymity of the collected data. Specifically, the questionnaire administered to parents consisted of questions about the number of cohabiting family members, the number of cohabiting and non-cohabiting brothers and sisters, citizenship, marital status, educational qualification, and employment status.

### Design of the Study

The questionnaire was designed on the basis of a research project agreed with the Presidency of the Council of Ministers of Italy, which co-financed the survey. The research design included the study of the following themes and social issues: quantity and quality of social interaction among peers; time spent in front of screens, cyberbullying, sexting and online grooming, psychophysical well-being, gender stereotypes and roles, and prosociality. Starting from the study of the scientific literature on social interaction, discomfort and deviance of the child population, the research group identified the specific dimensions of analysis as the quantity and quality of social interaction among peers, Internet use and online deviant behaviours, well-being and social attitudes and behaviours. The next phase concerned the operationalization of these dimension into variables able to measure the selected phenomena. This complex process led to the following main variables: vertical and horizontal social interactions (leisure, sports, and relational trust), online events and behaviours, electronic devices used and purposes and time of use, favourite video games, favourite social media, individual status in relation to the use of electronic devices, video games and social media, prosocial tendencies and behaviours, and adherence to gender roles.

To measure children’s levels of adherence to traditional gender roles, a list of actions and social roles was submitted to the interviewees asking who conducted them better with three options of answer: males, females or both [50,51]. From this list, two groups of questions were selected, composed of roles and actions related to the traditional female gender stereotypes, on the one hand, and to male gender stereotypes, on the other hand. Starting from these items, two indicators of adherence to female and male roles were constructed. For the construction of the indicator of female roles the chosen actions were the following: cooking, dancing, teaching, caring for children, cleaning, grocery shopping, talking for a long time on the phone and reading. For the construction of the indicator of male roles, the following actions and roles were selected: playing football, driving, being the leader at work, being the leader in the family, earning a lot of money, being president, playing video games, fighting in sports, being a scientist and working in police. Subsequently, a point was assigned to each response that identified actions or roles as exclusively with female for the first indicator and exclusively with male for the second. From the sum of the scores obtained, 4 levels of adherence to gender roles were defined: absent, low, medium and high, corresponding to the quartiles of the frequency distributions of responses. In addition, to investigate the influence of the family context on the internalisation of the gender roles of the participants, we built other two indicators concerning the level of education of the parents and their employment status. The parental educational level indicator was built on the parents’ educational qualification collected in the family socio-demographic form. Considering simultaneously the answers of both parents, 6 pre-coded modalities of response were synthesised to build an indicator consisting of 4 levels of parental education: low, medium-low, medium-high, and high. Finally, to simplify the reading of data, these 4 levels were synthetised in 2 levels of parental education: medium-low and medium-high. In the same way the parental employment status indicator was built, through the analysis of a question about the parental professional category of belonging. The outcome of the recoding process was a 4-level employment status indicator: low, medium-low, medium-high, and high. Additionally, in this case, a synthesis was carried out and the final indicator consisted in 2 levels of employment status: medium-low and medium-high.

In addition, to understand the relationships between the adherence to gender roles and children’s individual behaviours better, an indicator of prosociality was built. The prosociality indicator was based on three variables concerning the adoption of prosocial behaviours in different situations [52]. More specifically, participants were asked to express their opinion about: the utility of understanding the other’s feeling, their favourite way to compliment a friend, and how they try to understand what a friend feels. For each of these questions there were four possible answers linked to the presence of prosocial tendencies, a neutral behaviour, a self-centred tendency, and indifference. Analysing the collected answers, a single indicator of prosociality with 3 classes—low, medium, and high—to facilitate the bivariate analysis.

The process of socialisation in late childhood is also influenced by the strongly stereotyped contents of video games, social networks and applications [53,54,55]. In order to study the possible relationship between the screen time playing video games and using social media and applications and the internalisation of gender roles, two indicators were built. We used four variables investigating the time spent playing video games on the one hand, and that spent using social media and applications on the other hand, in terms of days per week and in hours per day. The results of the recoding process were the videogames screen time indicator and the social and app screen time indicator, identifying four use levels: absent, low, medium, high. Furthermore, in order to consider the total amount of time spent in front of a screen, another indicator was built synthesizing these two indicators and distinguishing four levels of screen-time: absent, low, medium, high.

The data analysis was conducted according to two steps: the first investigated the relationships between the indicators of adherence to male and female gender roles and the variables described in this paper, testing the independence of observations using the chi-squared test; the second step consisted in the attempt to observe these relations with a multivariate analysis using the binary logistic regression model. In this case, we used a synthetised version of the parental education indicator, parental employment status indicator and adherence to male and female gender roles indicators, consisting of two levels, low and high. To carry out these models, the forward method was used [56,57], in order to add gradually the significant variables and obtain the multiple dependence among the adherence to gender roles and the other individual and environmental variables.

## 3. Results

The male gender role indicator, as described in methodology, shows that 22.2% of respondents had a high level response to these stereotypes, 36.5% had a medium level, 27.1% had a low level, while only 14.3% of them did not have a stereotypical view of these social roles. Therefore, about half of the sample had medium and high levels of adherence to male gender roles (58%). However, there are important gender response differences. In fact, boys had a higher response level to male stereotypes than girls (26% boys and 18% girls) (Figure 1).

Similarly, the female gender roles indicator shows that 20% of respondents had a high response level to this stereotype, 33% had a medium level, and 29% had a low level. In this case, only 18% of them stated that these roles can be played by both men and women. Indeed, also for female roles, the percentage of adherence to the medium-high modalities was about half of the sample (53%) (Figure 2). However, observing the high modality, we find an inverted situation with respect to the acceptance of traditional male roles (Figure 1). The adherence to female gender roles is higher among girls rather than boys (23% of girls and 18% of boys). This first result shows how the internalisation of gender stereotypes is traceable already at this age, as boys and girls start identifying themselves with known behavioural models, reflecting the type of socialisation to which they have been exposed until that moment.

For the purposes of the study, we analysed the differences in the adherence to male and female gender roles in relation to other individual and environmental variables. The percentages of respondents who have internalised male (Figure 3) and female (Figure 4) gender roles were analysed in relation to the school grade, the district, the parental occupational status and the parental level of education. The age of participants follows a linear trend: the adherence to male gender roles decreased as the attended school grade of boys and girls increased. This is the only relation that can be traced between the adherence to traditional male roles and other individual and context variables, since the chi-squared test was significant.

The other relations with the context variables were not confirmed by the significance of the chi-squared test and therefore they could be of a spurious nature.

With regard to the adherence to female gender roles (Figure 4), there was not a linear relation with the age of the respondents. In this case, the chi-squared test was significant only for girls. This could indicate the different socialisation process to which boys and girls are exposed. For female students, in fact, the idea of the existence of a female social role of care and assistance could be more resistant, even with increasing age. In relation to the districts, this idea was more widespread among both boys and girls from schools located in the 6th district of Rome. This indicates the influence on these conditionings of the social context of belonging. The parental occupational status shows a similar trend for boys and girls, as the acceptance of female gender roles increases when the occupational status decreases for all the participants. Regarding the parental education indicator, the data do not show their influence on the phenomenon.

The percentages of respondents who have internalised male and female gender roles, combining medium and high levels of adherence in relation to the indicators measuring the levels of prosociality, the screen time on videogames and the screen time on social media and applications are shown in Figure 5 and Figure 6.

As the results show, the internalisation to males and females gender roles is higher for both boys and girls as the time they spent playing videogames increased. On the other hand, there are no differences in relation to the time spent on social media and applications. Concerning the relation between prosocial behaviour and the adherence to male gender roles among girls, it can be claimed that to a high level of prosociality correspond a lower level of adherence to these stereotypes. On the other hand, the adherence to female gender roles among boys decreased as the adoption of prosocial behaviour increased.

The relations examined so far in this paper were also found in the multivariate analysis. In fact, the model of binary logistic regression showed simple models, despite the adoption of a forward method that facilitates the insertion of variables for subsequent steps (Table 1). To carry out the four final models, we divided the sample by sex, using as dependent variables the two indicators about the adherence to male and female gender roles. The independent variables were the class attended, the district in which schools were located, the parental educational level, the parental occupational status, the level of prosociality and the screen time level. As expected from the considerations made about the bivariate analysis, the result showed that the only influential variables were the class attended, which is a proxy of the age of the respondents, and the level of prosociality.

## 4. Discussion

As the results show, the beliefs on the existence of roles and activities that should be carried out only by men or women are widespread among respondents. The percentage of children free from this social conditioning is 14% for male roles and 18% for female roles. Differences by sex in the internalisation of gender roles were showed, since their acceptance is higher among boys for male roles and among girls for female roles. These findings are in line with the results of previous studies, according to which boys and girls internalise more rigidly gender stereotypes associated with their own sex, as this schematisation of feminine and masculine traits allows them to develop their gender identity consistent with social expectations [26,58].

Furthermore, as highlighted before, the results show the presence of a linear relation between the class attended, which is a proxy of the respondents’ age, and the adherence to male roles for both boys and girls. In fact, in older children, the idea of activities and roles that are exclusively male is less spread than among younger children. This result confirms that late childhood is a delicate stage during which the immediate environment can make the difference in the internalisation of stereotypes, as along with a cognitive development we can observe a higher flexibility in the categorisation of the external world [59,60,61]. However, in relation to the adherence to female roles, we cannot find any relation with the age of the respondents. Looking at the older respondents attending the last year of primary school, more than half of the girls believes that some activities and roles can be performed or covered only by women. The reason could be linked to the persuasive power of sexist messages frequently coming from all learning environments that do not allow girls to reject the traditional idea of women with serious consequences on their aspirations and choices.

The attempt to find any relations between the endorsement of stereotypes and the parental educational level of respondents failed. In fact, the results of the bivariate analysis showed that the levels of adherence to traditional gender roles among children do not seem to be affected in an incisive way by the family and social contexts in which they live. The hypothesis that cultural and economic parental status, as well as the socio-economic environment play, at this age, a central role in determining the adherence to traditional male and female gender roles among children was falsified also by the results of the multivariate analysis, which showed the inconsistence of the relations among these variables.

However, the differences found in the percentages of adherence to traditional roles among children living in different family context are quite marked and they can be considered as indicative of trends that presumably will be found significant at older ages. Therefore, these differences could have a predictive role. These results can be explained by the fact that, in late childhood, these conditions are not yet stable, because they occur at different and non-standard times for each individual, making the sample under examination non-uniform with respect to the variables detected and therefore not suitable for a mathematical model, albeit probabilistic. Furthermore, children of this age are subject to evolutions linked to personal experiences and interactions within the social context they belong to. This can make it particularly difficult to grasp the influence of the cultural and economic status of the family at this age. It is important to underlie that these results could also be affected by the cultural involution linked to the spread of COVID-19, which has weakened the “protective” function that the level of education of parents should have. It can be said that the direct experience of situations of equality or disparity, which can derive from family habits and behaviours or from the social contexts, influence the creation of beliefs based on prejudices and stereotypes. Therefore, it is possible that these indicators about the surrounding environment of the children interviewed are not yet influential on this phenomenon due to the age of the respondents, which does not allow to find statistically significant relations. The reason could be that the process of the primary socialisation imbued with gender stereotypes still does not overlap with the secondary socialisation during which several factors, as peers, media but also the social context and education play a crucial role in determining the consolidation or the weakening of these social conditioning.

Today, the role of the media in influencing children’s opinions, beliefs and behaviours is increasingly important as videogames, social media and applications are a central part of children’s everyday lives. This became even more crucial considering the shift to a virtual sphere of our activities over the past two years due to domestic isolation during the spread of the COVID-19 pandemic. From stories and characters provided by tv programs, films and tv series, social media, YouTube or videogames, children can learn new things about gender and confirm or disprove their beliefs. According to our results, as the time spent playing with videogames increases, the adherence to female and male roles also increases both for girls and boys. This finding is not surprising since in videogames male and female characters are often shown in a hyper stereotyped way, with men depicted as strong heroes and women as attractive and sexy [53,62]. Previous studies have highlighted the significant relationship between the effect of the exposure to video games and the endorsement of sexist attitudes among adolescents [63,64]. Therefore, the effects of the continuous and prolonged exposure to stereotyped messages could be even more dangerous on the younger population since children have a more flexible mind and can be more easily persuaded by these distorted ideas about how men and women should appear. The time spent on social media and applications has made the difference only for the adherence to female traditional roles among the girls of our sample. This result suggests how the exposure to stereotyped female representations, which are very common on social media, not only can strengthen prejudices about female roles, but also become models to imitate for young girls [65].

Finally, the adoption of prosocial behaviours seems to inhibit the adherence to female roles, but only among boys. This seems to testify the existence of two distinct paths of socialisation, one male, which is oriented to power, and one female, which is oriented to care and family assistance. In other words, the more males believe in the existence of female social roles, the more the development of their prosociality could be inhibited.

## 5. Conclusions

The analyses carried out in this study reveal that gender stereotypes are widespread among children aged 8–11, as the adherence to traditional gender roles was around 60%, regardless of the age and sex of respondents. Unlike other older population groups, it is present in childhood for cultural reasons and independently of family and contextual variables. In fact, at this age, the mediation of secondary socialization agencies on the development of gender identity, such as the influence of school, is still weak, and children have few direct experiences other than family ones.

However, it is important to underline that the survey was carried out during the spread of the COVID-19 pandemic, when the exposure to the family context has strengthened due to the long periods of domestic isolation. The trends presented in this paper are particularly important and alarming if we consider that the family is the main theatre of the reproduction of social inequalities, but also taking into account the triggering of a cultural regression that has reinforced the adhesion and introjection of stereotyped ideas during the lockdown period. In fact, as some studies have shown, domestic confinement has strengthened the idea of rigid gender roles, favouring the reliance on the traditional family model, according to which only women are assumed to perform housework and to take care children [15,16,17]. This backwards step in terms of gender equality may have affected also attitudes and opinions of children as they learn from what they see in their immediate environment, which mostly has coincided with their family during the last two years.

The data reveal that the first elements of adherence to gender roles and the consequent adoption of stereotyped behaviours and attitudes is still at the first stage, to the point that age is crucial for the understanding of the phenomenon. The gender differences that have been described also illustrate a picture that shows how at this age the gap between male and female behaviours widens. Indeed, for girls the adherence to gender roles is linked to age, while boys are not yet affected by their own growth, showing different levels of adherence to gender roles in relation to their level of prosociality, which acts as a mitigating and protective element. However, it is important to highlight that prosociality is a complex phenomenon since the development of prosocial behaviours begins in preschool age, and during late childhood, it is already affected by the socialisation process, which reduces the influence of its biological matrix. In fact, as the age increases, the tendency to prosociality becomes increasingly more the result of the type of socialisation to which people are exposed [66].

The success of a survey carried out on a sample basis, through an assisted administration of a structured questionnaire composed of many variables, was also in outlining the effects of socialisation (in particular the primary one) on the development of beliefs and attitudes that will affect the future choices of young people, as well as the future levels of social inclusion and exclusion. On the other hand, this research foreshadows the importance of extending this kind of studies on a wider geographical scale. This will allow to fill the information gap on this age group, due to the scarcity of sample surveys, which are particularly difficult due to the age of the respondents, and to verify the trend of the observed phenomenon in different social contexts. The production of periodically, verifiable and repeatable information about this age group is undoubtedly useful also in order to make appropriate diachronic comparisons. Furthermore, due to the psychosocial effects of the COVID-19 pandemic, the role of hyperconnection should be strongly taken into account, since it has increased during the last two years. Since the regression of this phenomenon to the pre-pandemic levels of exposure to social media seems highly unlikely, the effects of the increase in the screen time could be measured in the near future. The over-exposure to virtual or family environments imbued with stereotypes, could accelerate and exacerbate the sedimentation of ideas conforming to traditional and rigid gender roles.

The development of this phenomenon during late childhood should be observed carefully in the future, analysing also its trends on pre-adolescents and adolescents. Unmasking and rejecting the spread of social conditionings through targeted and effective interventions, based on an increasingly precise and comparative knowledge, is a precondition for creating a society increasingly free from discrimination and violence. It can be said that up to the age of 11 boys and girls can be easily influenced by the stimuli of their immediate environment, because the individual gender schemes of children are still under development. For this reason, the implementation of actions promoting a personal development free of social constraints and mental barriers could be even more effective than actions targeting older students.

Although there are several programs that address gender stereotypes aimed at both families and teachers, they are not implemented on a large scale, and little is known about their effectiveness. A virtuous example comes from Sweden, which has developed gender-neutral schools, where teachers pay attention not only to the activities carried out, but also to the language used. The results of a study that compared the behaviour of children attending these gender-neutral schools and that of children attending traditional schools found that the first group was less stereotyped and also more interested in relating to children of the opposite sex and to strangers [67]. More egalitarian and gender-neutral school environments can make a difference in creating safe and inclusive spaces also for gender-variant children.

The implementation of interventions that reduce the impact of a stereotyped socialisation is even more urgent today, due to the cultural regression which, as future studies could be confirm, was triggered by the pandemic.

## Figures and Tables

**Figure 1 ijerph-19-03408-f001:**
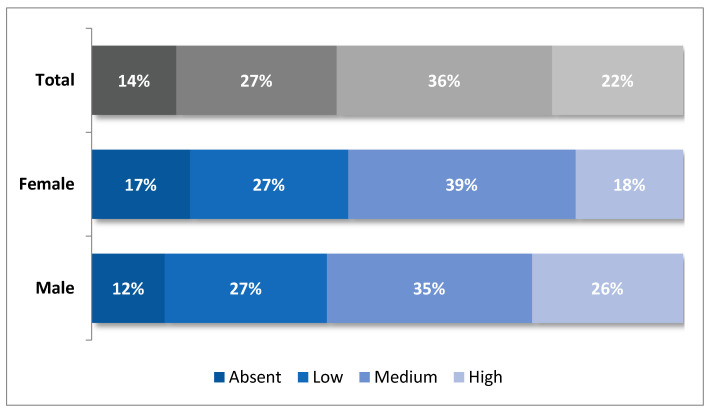
Levels of adherence to male gender roles (% by sex).

**Figure 2 ijerph-19-03408-f002:**
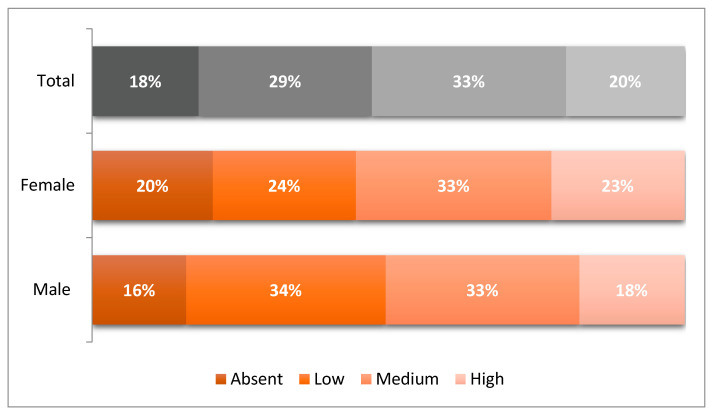
Levels of adherence to female gender roles (% by sex).

**Figure 3 ijerph-19-03408-f003:**
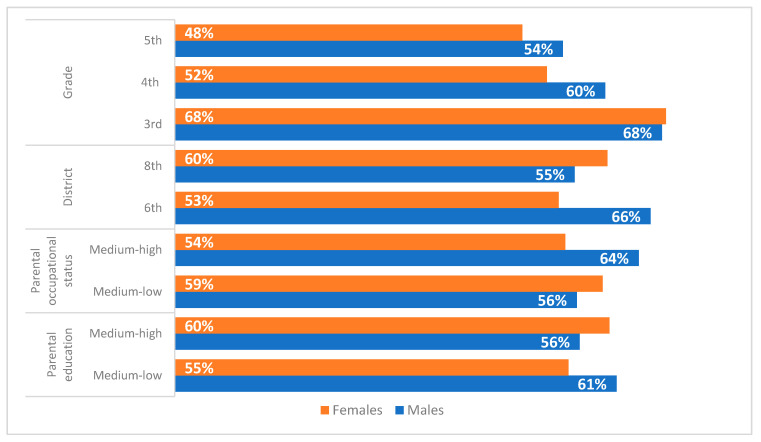
Levels of adherence to male gender roles by sex (% of medium and high levels by school grade, district, parental occupational status and parental level of education).

**Figure 4 ijerph-19-03408-f004:**
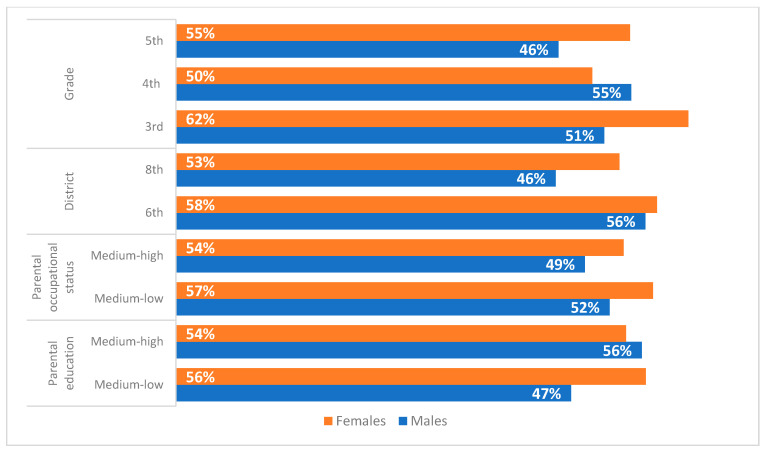
Levels of adherence to female gender roles by sex (% of medium and high levels by school grade, district, parental occupational status and parental level of education).

**Figure 5 ijerph-19-03408-f005:**
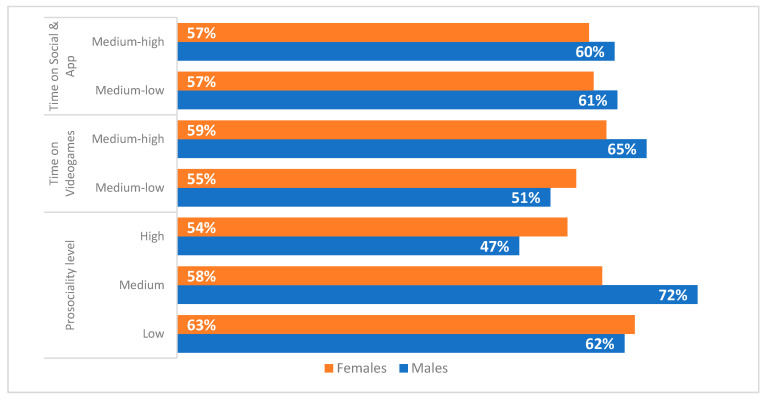
Levels of adherence to male gender roles by sex (% of medium and high levels by prosociality, screen time playing videogames and screen time spent on social media and applications).

**Figure 6 ijerph-19-03408-f006:**
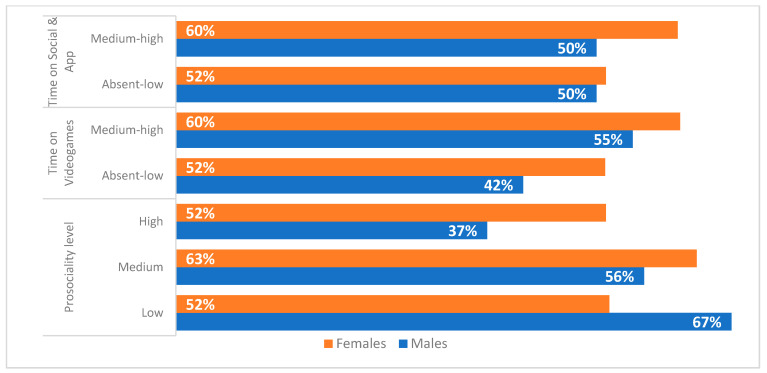
Levels of adherence to female gender roles by sex (% of medium and high levels by prosociality, screen time playing videogames and screen time spent on social media and applications).

**Table 1 ijerph-19-03408-t001:** Results of binary logistic regression models.

Male Gender Roles							
Sex		B	S.E.	Wald	df	Sign.	Exp(B) or OR
**Female**	**Constant**	1.183	0.415	8.127	1	0.004	3.266
	**Grade**	−0.439	0.197	4.986	1	0.026	0.644
**Male**	**Constant**	1.198	0.387	9.591	1	0.002	3.312
	**Prosociality**	−0.353	0.143	6.107	1	0.013	0.702
**Female gender roles**							
**Sex**		**B**	**S.E.**	**Wald**	**df**	**Sign.**	**Exp(B) or OR**
**Female**	**Constant**	0.245	0.157	2.427	1	0.119	1.278
**Male**	**Constant**	0.893	0.392	5.195	1	0.023	2.443
	**Prosociality**	−0.392	0.148	7.008	1	0.008	0.676

B—Coefficient model. S.E.—Standard error around the coefficient B. Wald and Sig.—This is the Wald chi-square test that tests the null hypothesis that the constant equals 0. This hypothesis is rejected when the *p*-value (listed in the column called “Sig.”) is smaller than the critical *p*-value of 0.05 (or 0.01). Usually, this finding is not of interest to researchers. For that we have reported only the results for *p*-values < 0.05. df—This is the degrees of freedom for the Wald chi-square test. There is only one degree of freedom because there is only one predictor in the model, namely the constant. Exp(B)—This is the exponentiation of the B coefficient, which is an odds ratio (OR). Odds ratios can be easier to interpret than the coefficient, which is in log-odds units.

## Data Availability

Data of this study are not available because of the co-property with the Italian Presidency of the Council of Ministers.

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
