# Peer review of "Are We Still a Sexist Society? Primary Socialisation and Adherence to Gender Roles in Childhood"

_ijerph, 2022, doi:10.3390/ijerph19063408_

Round 1
Reviewer 1 Report
This paper addresses an important topic area relative to gender roles and how they are internalized and replicated by children, producing an adherence to traditional gender stereotypes. The authors focus on primary school age children and demonstrate through the data and data analysis that children do significantly internalize gender stereotypes and this potentially serves to perpetuate gender inequality. It may be helpful in the discussion to refer readers to programing that seeks to disrupt the internalization of gender stereotypes as this seems to be important in the argument that gender inequality is tied to early internalization of gender differences. The paper is well written and relative to current discussions about gender inequality. It may be worth mentioning how the internalization of stereotyped gender roles impacts how gender is expressed by those who are gender non-conformist, e.g., gender queer. In some ways, the conclusions of this paper hint at re-imagining gender and perhaps promoting non-traditional gender expression.
Author Response
Dear reviewer,
The study underlies that during primary socialisation the most important figure is the family but also the social context in which children grow up is crucial for their education. If adults act in a gender-neutral manner by avoiding categorisations among the roles played by men and women, children would also have the advantage of starting social life without prejudices. This attitude cannot be achieved in a short time but it must be constantly stimulated for a social evolution in the sense of gender neutrality. In the conclusion paragraph a section about a virtuous program developed in Swedish schools has been added, with a reference to gender variant children.
Reviewer 2 Report
Review sexist socialisation
Are We Still A Sexist Society? Primary Socialisation and Adherence to Gender Roles in Childhood
This article argues for the importance of representative surveys of gender stereotypes in populations aged between 8-11, because gender roles are learnt but malleable at this developmental stage, and because they have important implications for future opportunities as well as overall societal gendered inequalities. It reports a survey conducted on primary school children in Rome in 2021, designed to offer an objective of measurement of adherence to gender roles among children aged 8-11 in that city, and to explore any links between gender role internalisation, parental education levels (measured by occupation), and SES (measured by the location of the school?), and other factors such as playing computer games. It aimed to ask: what is the adherence to gender roles in children, are there statistical differences in how male and female children adhere to their associated gender roles, does adherence to gender roles interact with a range of other relevant factors (quantity and quality of social interaction among peers; time spent in front of screens (screen time), cyberbullying, sexting and online grooming, psychophysical well-being, prosociality); in addition, it aimed to address the hypotheses that higher levels of parental education and more advantage socioeconomic contexts would be correlated with lower levels of adherence to traditional gender roles. Key findings of the study include that, just over half the sample scored moderately or high adherence to their respective gender role. Prosociality and time spent playing videogames were also identified as having important relationships to adherence to traditional gender roles and that being prosocial was negatively correlated with the male role. Sex differences were identified in terms of how age mediated adherence to gender role stereotypes.
The study offers an interesting method and insight into an important issue. I, therefore, recommend it for publication, subject to the authors addressing the issues I raise below.
Abstract
The authors need to say where the data was collected, although the main document highlights the importance of contextualising the research, the abstract talks about “children” in an inappropriate universalising way.
line 17, “ while any significant relationship was found” is there a missing ‘no’?
Results need to describe what was found without interpretation. For example the statement ‘
Due to a distinct path of primary socialisation, differences by sex have been found’ includes interpretation in relation to a distinct path of primary socialisation, which has not been measured in the study and so this is an interpretation of the findings rather than results. I recommend the authors go to their results section again and identify their key findings and more clearly summarise these in the abstract. The conclusion section of the abstract should start with the main interpretations of the findings, addressing research questions and hypotheses, and then discuss implications for future research.
The literature review was clear and succinct. However, it might be developed with a short paragraph explaining the history of how to measure gender role stereotypes, eg Bem’s sex role inventory. This addition would then allow researchers to make a case for why it was important to develop their own measure. (Then in the method section, they can explain the process and rationale for developing their survey).
Further, prosociality is an important measure in the survey, but the literature does not discuss this topic or why it is important for the study. So a section on why prosocial measures are important for gender stereotype studies needs to be added to the literature review. And perhaps a brief discussion on the other measures that were considered to be important, what is the literature underpinning why they would be included for consideration in studies on adherence to gender roles?
Recommend the literature section finishes with a clear set of statements describing all the research questions and hypotheses (currently only some given).
Method section
For clarity, this section could be rewritten using more subheadings, or at least, more paragraphs to distinguish key elements of the method. It also needs an additional section on developing the survey. Details as to how the data were collected are given, but it’s unclear how or why the survey was designed. Why not use an existing validated survey? If these do not exist or are not relevant for the current context, they need to explain this and offer a rationale for the content of the survey that was used. For example, where did the 42 questions come from and what thinking/theory/logic/testing underpinned their development? An explicit section on “design of the survey” is therefore needed and have this separate to the section on how the authors analysed the survey. In the ‘design of the survey’ section the authors can also add information about why the other measures were used, to provide an explanation for covering quantity and quality of social interaction among peers; time spent in front of screens (screen time), cyberbullying, sexting and online grooming, psychophysical well-being, and prosociality before discussion of how they were analysed (ie discussion of these variables which happens on lines 176+ could be moved to an earlier, a new section that focuses on the rationale for the survey design).
Results
I raise for the authors' attention consideration of bias in how they communicate the findings, which might lead to overemphasising findings. For example, saying something is more than half the sample when it’s only 53% (ie 3% over half) (see line 210). Given that the authors say that Italy is highly gendered stereotypes country, perhaps it’s also interesting to highlight that just under half had absent or low adherence to gender roles, which perhaps create some more hopeful interpretation of the data and currently being offered?
Given the traditional distinction of results sections for reporting findings and discussion sections for interpreting them; I recommend that the authors take the interpretation of the data out of the results section and move it into the discussion section. This would make the results section clearer and more streamlined in describing the findings (without interpretation). I also recommend organising them more clearly in relation to the research questions and hypotheses stated at the end of the introduction (as recommended above).
Discussion of how non-significant findings might actually be significant (approx line 231) could be moved to the discussion since this involves significant interpretation. And perhaps toned down, they weren’t significant. Similarly, interpretation of the findings from line 246 should be moved to the discussion.
Lines 266+ is unclear. if adherence to female gender roles decreases as pro-social behaviour increases this would imply that male gender roles are connected to prosocial behaviour, and that’s not what is being argued in this paragraph.
Discussion section
Recommend the discussion starts with the first/central research question, namely levels of adherence to gender roles, taking some of the discussion around answering that question that is currently in the analysis section and putting it here. This could then be followed by interpretation of the other research questions/hypotheses about what factors might predict this adherence (i.e. parental status, age etc.)
Overall writing style
Some slippage in terminology, for example, the method is described as ‘interviews’ but I think it’s more appropriate to call them surveys?
Final read through needed e.g., the first sentence in the introduction is unclear, as is the sentence starting Line 243, and sentence starting Line 261. Very long paragraphs, especially in the method section. Stay in the past tense e.g. the method flips between the present and past tense sometimes. Typos in lines 12, 208, 258.
Author Response
Dear reviewer,
The study underlies that during primary socialisation the most important figure is the family but also the social context in which children grow up is crucial for their education. If adults act in a gender-neutral manner by avoiding categorisations among the roles played by men and women, children would also have the advantage of starting social life without prejudices. This attitude cannot be achieved in a short time but it must be constantly stimulated for a social evolution in the sense of gender neutrality. In the conclusion paragraph a section about a virtuous program developed in Swedish schools has been added, with a reference to gender variant children.
We respond point by point to the review with an attached file
